# Umbilical Cord SFRP5 Levels of Term Newborns in Relation to Normal and Excessive Gestational Weight Gain

**DOI:** 10.3390/ijms20030595

**Published:** 2019-01-30

**Authors:** Żaneta Kimber-Trojnar, Jolanta Patro-Małysza, Marcin Trojnar, Dorota Darmochwał-Kolarz, Jan Oleszczuk, Bożena Leszczyńska-Gorzelak

**Affiliations:** 1Chair and Department of Obstetrics and Perinatology, Medical University of Lublin, Lublin 20-090, Poland; jolapatro@wp.pl (J.P.-M.); jan.oleszczuk@umlub.pl (J.O.); b.leszczynska@umlub.pl (B.L.-G.); 2Chair and Department of Internal Medicine, Medical University of Lublin, Lublin 20-081, Poland; marcin.trojnar@umlub.pl; 3Department of Gynecology and Obstetrics, Institute of Clinical and Experimental Medicine, Medical Faculty, University of Rzeszow, Rzeszów 35-959, Poland; ddarmochwal@ur.edu.pl

**Keywords:** adipokines, secreted frizzled-related protein 5, leptin, ghrelin, excessive gestational weight gain, neonatal anthropometry, obesity

## Abstract

Among the new adipokines, secreted frizzled-related protein 5 (SFRP5) is considered to prevent obesity and insulin resistance. The umbilical cord SFRP5 levels have not yet been investigated. The main aim of the study was to investigate whether the umbilical cord SFRP5 concentrations are altered in term neonates born to mothers with excessive gestational weight gain (EGWG). Two groups of subjects were selected depending on their gestational weight gain, i.e. 28 controls and 38 patients with EGWG. Umbilical cord and maternal serum SFRP5 levels were lower in the EGWG group. Umbilical cord SFRP5 concentrations were directly associated with the maternal serum SFRP5, hemoglobin A1c and lean tissue index, umbilical cord leptin levels, as well as newborns’ anthropometric measurements in the EGWG subjects. In multiple linear regression models performed in all the study participants, umbilical cord SFRP5 concentrations depended positively on the maternal serum SFRP5, ghrelin, and leptin levels and negatively on the umbilical cord ghrelin levels, low-density lipoprotein cholesterol, pre-pregnancy body mass index, and gestational weight gain. EGWG is associated with disturbances in SFRP5 concentrations. Obstetricians and midwives should pay attention to nutrition and weight management during pregnancy.

## 1. Introduction

As a novel adipokine mainly secreted from the adipose tissue, secreted frizzled-related protein 5 (SFRP5) contains a cysteine rich domain as well as a netrin-like function domain, and it plays a regulatory role in the wingless-type Mouse Mammary Tumor Virus (MMTV) integration site family member (Wnt) signaling pathways [1,2,3]. Preliminary clinical and basic research reveals that the biologic function of SFRP5 may be similar to adiponectin, which exerts an anti-inflammatory effect in the metabolic homeostasis [1,2]. SFRP5 has been reported to be implicated in obesity, insulin resistance, dyslipidemia, and metabolic syndromes [4,5,6,7,8,9,10]. Circulating concentrations of SFRP5 have been measured in healthy and diseased individuals in several studies, but there are still limited data concerning SFRP5 in obstetric aspects [3,11,12,13,14].

As far as we know, there is no reported study investigating the SFRP5 concentrations in the human umbilical cord blood. We hypothesized that SFRP5 concentrations would probably be impaired in the umbilical cord of full-term neonates born to excessive gestational weight gain (EGWG) mothers. The aim of this study was also to investigate whether the umbilical cord SFRP5 levels correlate with selected maternal parameters and neonatal anthropometric measurements.

## 2. Results

Compared with the healthy study participants, the EGWG mothers had comparable age and pre-pregnancy BMIs, but they presented significantly higher BMIs at and after delivery. The EGWG women were also characterized by increased levels of hemoglobin A1c (HgbA1c), triglycerides and indexes of fat (FTI) and lean (LTI) tissues as well as lower concentrations of high-density lipoprotein cholesterol (HDL). The maternal SFRP5 levels were decreased in the serum in the EGWG group. Lower SFRP5 concentrations as well as higher ghrelin and leptin levels were observed in the umbilical cord blood of neonates born to the EGWG mothers. No significant differences were noticed between the groups with regard to other analyzed parameters, including the maternal serum ghrelin and leptin levels as well as neonatal anthropometric measurements (Table 1).

In the control group, the umbilical cord SFRP5 concentrations correlated positively with the maternal serum HgbA1c, SFRP5, ghrelin and leptin levels, and the neonatal chest circumference. We found negative correlations between the umbilical cord SFRP5 and BMIs (pre-pregnancy, at and after delivery), total cholesterol, and umbilical cord ghrelin levels in the control subjects (Table 2).

In the EGWG group, we observed a direct correlation between the umbilical cord SFRP5 and the maternal serum HgbA1c, SFRP5 and LTI after delivery, the umbilical cord leptin levels, and all four newborns’ anthropometric measurements (i.e. with neonatal birth weight, birth body length, and head and chest circumference). Negative correlations were revealed between the umbilical cord SFRP5 concentrations and gestational weight and BMI gains, albumin, total cholesterol, HDL, and the umbilical cord ghrelin levels in the EGWG subjects (Table 2).

In multiple linear regression models performed in all the study participants, after adjustment for the maternal serum SFRP5 levels, the serum and umbilical cord ghrelin and leptin levels, maternal low-density lipoprotein cholesterol (LDL), triglycerides, HgbA1c, gestational weight gain, pre-pregnancy BMI, BMI at delivery and gestational BMI gain, we noted that the umbilical cord SFRP5 concentrations were positively dependent on the maternal serum SFRP5, ghrelin and leptin levels as well as negatively dependent on the umbilical cord ghrelin levels, LDL, pre-pregnancy BMI and gestational weight gain (Table 3).

The Benjamini–Hochberg correction for false positive results revealed that all of the originally significant associations were still significant.

## 3. Discussion

We decided to choose EGWG and not pre-pregnant obese women, as EGWG is mainly linked to overnutrition during a relatively short period of time (with regard to life expectancy), i.e. within the last nine months. Gestational weight guidelines of the Institute of Medicine (IOM) [15] provide ranges of recommended weight gain for specific pre-pregnancy body mass index (BMI) categories in relation to the least risk of adverse perinatal outcomes. It is recommended that in order to prevent adverse maternal as well as infant outcomes, women with normal weight at the time of conception should limit their total weight gain in pregnancy to 11.5–16 kg, overweight women to 7–11.5 kg, and obese women to 5–9 kg [15]. Goldstein et al. revealed in a systematic review of 23 cohort studies in 1.3 million women that 47% of women exceeded the upper limit of IOM-recommended weight gain [16]. EGWG, which is usually due to improper nutrition during the pregnancy period, has been regarded as a potentially modifiable, independent risk factor not only for the development of maternal overweight and obesity but childhood adiposity as well [17,18]. EGWG may expose the developing fetus to persistently raised concentrations of glucose, insulin, amino acids, and lipids as well as imbalance between pro- and anti-inflammatory adipokines derived from maternal adipose tissue [19,20].

SFRP5 is an anti-inflammatory adipokine that regulates metabolic homeostasis [5,21]. The classical molecular mechanism of SFRP5 is designated to inhibit the combination of Wnt protein with its cell membrane receptors (frizzled protein) and block the downstream Wnt signaling pathways through binding with the extracellular Wnt-5a or Wnt-3a [2,22,23]. *Sfrp5* knockout mice fed a high fat diet developed adipose macrophage infiltration, severe glucose intolerance, and hepatic steatosis [1,2,24].

SFRP5 is an inhibitor of Wnt signaling, the crucial signaling pathway in the placental vascular development. Placental angiogenesis is a pivotal process that establishes feto-maternal circulation, ensures efficient materno-fetal exchanges and contributes to the overall development of the placenta throughout pregnancy. Any failure in these processes will definitely result in the development of many gestational complications such as preeclampsia, GDM, and intrauterine growth restriction [25,26,27]. Nevertheless, there are limited data concerning SFRP5 in the obstetric aspects. A previous study demonstrated that first trimester serum SFRP5 levels were significantly lower in the pregnant women who subsequently developed GDM in comparison to the healthy pregnant women [3]. Based on the mechanism that SFRP5 is an inhibitor of the Wnt signaling pathway, which is implicated in the regulation of insulin resistance, inflammation, and placental vasculature, it was suggested that altered levels of SFRP5 may contribute to the development of GDM [3,28,29].

It is worth highlighting that some of the previous studies evaluating the levels of various adipokines in the umbilical cord blood did not take into consideration the maternal BMI and weight gain during pregnancy [30,31,32,33]. Our study comprised both participants with normal pre-pregnancy BMI and different gestational weight gain; i.e. an excessive increase in body weight during pregnancy in the EGWG group and an appropriate gestational weight gain in the control group and their offspring. Our results revealed differences at the periparturient period not only in the gestational weight and BMI gains between these two groups of mothers but in the laboratory results as well. Lower levels of HDL as well as higher HgbA1c and triglycerides concentrations were present in the EGWG group. It seems that in many ways the pregnancies of women with EGWG resemble pregnancies complicated by gestational diabetes mellitus (GDM). What is important in this context is that the women with a history of GDM exhibit altered risk factors of cardiovascular diseases, including lower HDL concentrations, when compared with mothers with healthy pregnancies [34,35,36].

Nonetheless, the offspring of our control and EGWG groups had comparable anthropometric measurements, including birth weight. However, in light of the previous studies, the fetal metabolic programming may occur within normal birth weight ranges [37,38]. Lawrence et al. [39] reported a potential role of genetics for the existing association between the maternal weight gain during pregnancy and offspring BMI change. The association between the gestational weight gain and offspring BMI change attenuated by 28% when a genetic score was added, but the offspring’s genetic variation did not play a role in the association. The authors speculate that epigenetics may underlie this finding [40]. DNA microarray analyses on the placental junctional zones performed by Gao et al. [13] revealed that the *Sfrp5* expression was decreased in the rats fed a low protein diet, which may activate non-canonical Wnt signaling. Christodoulides et al. [41] reported that *Sfrp5* mediated epigenetic silencing of the Wnt signaling pathway in the white adipose tissue could lead to an increased adipogenesis with a significant likelihood of increasing susceptibility to diet-induced obesity in the mice models. In addition, SFRP5 has been demonstrated to inhibit the activation of c-JunN-terminal kinase (JNK) downstream of the Wnt signaling pathway [1,5,42].

To the best of our knowledge, this study is the first report evaluating the umbilical cord SFRP5 in the offspring of healthy mothers as well as of women with excessive gestational weight gain. We also investigated associations between its levels in the umbilical cord and in the maternal serum. Our study revealed that the umbilical cord SFRP5 levels were lower in the offspring of the EGWG mothers. However, what is also extremely important is that the studied mothers with EGWG, when compared with the healthy controls, presented lower SFRP5 concentrations in the serum. The umbilical cord SFRP5 levels were positively associated with the maternal serum SFRP5 levels in both groups. Apart from this, we performed multiple linear analyses that revealed the dependence of the umbilical cord SFRP5 concentrations on its levels in the maternal serum. Each 1 ng/mL decrease in the maternal serum SFRP5 concentration was associated with a decrease in the umbilical cord SFRP5 level by 0.33 ng/mL.

The study of Prats-Puig et al. [43] reported concomitantly decreased concentrations of SFRP5 in obesity markers of prepubertal children. The cited authors found that lower serum SFRP5 potentiated the association between Wnt-5A and insulin resistance. Previous studies showed that circulating SFRP5 was decreased in patients with impaired glucose tolerance or T2DM and was associated with various obesity-related metabolic parameters [8,9,43]. SFRP5 sequesters Wnt-5A, thereby attenuating the activation of c-Jun N-terminal kinase 1 [43,44,45]. Prats-Puig et al. [43] concluded that SFRP5 may be an anti-inflammatory adipokine that could be negatively regulated during obesity development, leading to a less-favorable metabolic phenotype. Their results also suggested that a failure to upregulate SFRP5 in obesity may lead to unrestrained pro-inflammatory actions of Wnt-5A, resulting in metabolic dysfunction [1,43,46].

The expression of *Sfrp5* is slowly induced in the process of differentiation of white and brown adipocytes and increased in mature adipocytes [47]. Lower SFRP5 levels have been detected in obese subjects in contrast with lean subjects in studies analyzing correlations between SFRP5 and adiposity indicators such as BMI, waist–hip ratio, body fat percentage, and lipid profile [9]. In our study, a positive correlation was found between the umbilical cord serum SFRP5 concentrations and maternal LTI, but only in the EGWG group. LTI, which is defined as the lean tissue mass divided by the square of the body height, expresses the muscle mass [48]. Mori et al. [49] revealed that SFRP5 promotes adipocyte growth by repressing Wnt signaling and decreasing oxidative metabolism as an endogenous suppressor of adipogenesis. Rulifson et al. [50] and Van Camp et al. [51] found that a significant increase in *Sfrp5* expression was observed amongst adipose tissues in obese mice and that genetic variation in *Sfrp5* could determine the distribution and volume of both subcutaneous and abdominal fat in obese males, respectively.

As far as progressive metabolic complications are concerned, it should be pointed out that SFRP5 levels decreased with age in the prepubertal children [43]. On the other hand, increased leptin concentrations during puberty were found to be a reliable indicator of insulin resistance associated with increasing age [52]. Thus, the possibility that reduced SFRP5 levels may contribute to the state of insulin resistance associated with increasing age and pubertal development cannot be excluded [43,53].

In this study, we evaluated the concentrations of leptin and ghrelin, which were higher in the umbilical cord blood of EGWG patients than in the control subjects. However, concentrations of the maternal serum ghrelin and leptin were similar in both studied groups. The previous results suggested that ghrelin may play a role in the fetal adaptation to intrauterine malnutrition [54]. Moreover, it is interesting to point out a negative correlation between SFRP5 and ghrelin concentrations in the umbilical cord of both the control and EGWG newborns. What is more, we were also able to find a positive association between the umbilical cord SFRP5 and leptin levels but only in the EGWG group.

We observed that the SFRP5 umbilical cord levels were associated with the maternal BMI values. We found out that the umbilical cord SFRP5 negatively correlated with the maternal BMIs (pre-pregnancy, at and after delivery) in the control subjects as well as with the gestational weight and BMI gains in the EGWG group. Besides, multiple linear regression models performed in all the study participants revealed that the umbilical cord SFRP5 concentrations were negatively dependent on the pre-pregnancy BMI and gestational weight gain. It is noteworthy that every 1 kg of gestational weight gain was linked to a decrease in the umbilical cord SFRP5 concentration by 0.08 ng/mL. Similarly, we made an observation that the umbilical cord SFRP5 levels correlated with the maternal concentrations of total cholesterol and HgbA1c and were negatively dependent on LDL. These results seem to confirm the impact of maternal weight and metabolic imbalance on SFRP5 in newborns. We cannot relate our results to the previous ones because no such results are available. However, a comparison with the observations made in non-pregnant humans concerning the serum SFRP5 levels negatively associated with BMI values [3,8,9] as well as with insulin resistance and lipid profile [5,9,28] might confirm the existence of a feto-maternal unit and fetal dependence on the maternal nourishment status.

Our study methodology relied on an accurate selection of the study subjects. We decided to choose EGWG and not pre-pregnant obese women. In the case of pre-pregnancy obesity the analysis of the results would have to take into consideration the influence of modulators such as dyslipidemia, hypertension, insulin resistance, pre-pregnancy treatment of obesity, and disorders of the carbohydrate–lipid balance. Because we chose the EGWG group, we were able to reduce the number of interfering and confounding factors in the analysis of the study results. The study groups were formed on the basis of women’s similar age, normal pre-pregnancy BMIs, and term pregnancies. Selected women were to be free of any chronic and gestational diseases and receive only vitamins throughout their pregnancy period. On the other hand, this study has an important limitation as a small sample study. Our results require further verification. It also appears of great clinical significance to monitor the circulating SFRP5 levels in all trimesters of pregnancies as well as in future life of children.

## 4. Materials and Methods

The study comprised infants of mothers who were in a singleton term pregnancy (after 37 weeks of gestation) and delivered at the Chair and Department of Obstetrics and Perinatology, the Medical University of Lublin. The data collection was performed between March 2016 and February 2017. All the study subjects included in this study were Caucasian. The study included women receiving only vitamin-iron supplementation during pregnancy, without any exclusion criteria given below, with normal pre-pregnancy BMI values and three normal results of the two-hour, 75 g oral glucose tolerance test at 24–28 weeks of gestation [55,56]. Depending on the achieved gestational weight gain, two groups were strictly selected:healthy controls—28 pregnant women with normal gestational weight gain;patients with EGWG—38 pregnant subjects with excessive gestational weight gain.

Exclusion criteria from the study were as follows: multiple pregnancy, chronic infectious diseases, urinary infections, anemia, metabolic disorders (except for improper gestational weight gain for the EGWG group), mental illness, cancer, liver diseases, cardiovascular disorders, fetal malformation, premature membrane rupture, and intrauterine growth retardation. We had to rule out, among others, all females with gestational hypertension, which is a common complication observed in patients with high BMI values in the third trimester of pregnancy.

Anthropometric measurements of the mothers were performed immediately before and after delivery. Calculation of pre-pregnancy BMIs values were based on body weight measured at the first prenatal visit, occurring in the first trimester (before 10 weeks of gestation). We defined total gestational weight gain as the difference between the mother’s weight at delivery and her pre-pregnancy weight. We calculated gestational BMI gain as well. Neonatal anthropometric measurements, including birth weight, body length, and head and chest circumferences, were performed immediately after birth. The maternal serum levels of albumin, HgbA1c, and lipid profile were measured at a certified laboratory. The cord blood samples were taken during delivery but without any interference with its course. The maternal serum samples were taken after delivery, taking into account a 6 h fasting period. After centrifugation, all the collected cord blood serum as well as maternal serum samples were stored at −80 °C. Concentrations of SFRP5 and ghrelin in these materials were determined using commercially available kits and in compliance with the manufacturer’s instructions (Wuhan EIAab Science Co., Wuhan, China) via traditional enzyme-linked immunosorbent assay (ELISA). Concentrations of leptin in the cord blood serum and maternal serum were determined using commercially available kits and in compliance with the manufacturer’s instructions (R&D Systems, Inc., Minneapolis, MN, USA) via ELISA. The survey was performed in duplicates for each patient.

Maternal body composition (LTI and FTI) were evaluated by the bioelectrical impedance analysis (BIA) method with the use of a body composition monitor (BCM) (Fresenius Medical Care) in the early post-partum period (i.e. 48 h after delivery).

All the patients were informed about the study protocol and a detailed written consent was obtained from each patient who agreed to participate in the study. A separate information sheet was prepared for the parents of newborns. Written signed consent from each infant’s legal guardians (mothers) was obtained after informed consent.

The study protocol received approval of the Bioethics Committee of the Medical University of Lublin (no. KE-0254/221/2015 (25th June 2015) and no. KE-0254/348/2016 (15th December 2016)).

All values were reported as the median (interquartile range 25–75%) or numbers and percentages. Differences between groups were tested for significance using the Mann–Whitney *U* test. The Spearman’s coefficient test was used for the correlation analyses. Categorical data were compared using the chi-square test. The Benjamini–Hochberg correction for false positive results was performed. Multiple linear regression model was used to examine the association between the umbilical cord SFRP5 levels and the selected biophysical and biochemical parameters of the mothers and their offspring. Regression models were adjusted for the serum SFRP5 levels, the serum and umbilical cord ghrelin and leptin levels, LDL, triglycerides, HgbA1c, gestational weight gain, pre-pregnancy BMI, BMI at delivery, and gestational BMI gain. All analyses were performed using Statistical Package for the Social Sciences software (version 19; SPSS Inc., Chicago, IL, USA). A *p*-value of <0.05 was considered statistically significant.

## 5. Conclusions

In light of our study results, it becomes understandable that maternal condition, including gestational weight gain, should be of utmost importance when doing further research into the umbilical cord SFRP5 concentrations and investigating their relationship with the fetal and neonatal anthropometry and metabolic state. Evaluation of the umbilical cord SFRP5 levels in the offspring of EGWG mothers as well as of the relationships between the umbilical cord SFRP5 levels, maternal laboratory results, and neonatal anthropometric measurements is a new idea.

It is also quite obvious that obstetricians and midwives should pay attention to nutrition and weight management of their pregnant patients and should educate them on the dangers of overnutrition and on how it predisposes not only them but also their children to metabolic disorders later in life. On the other hand, it should be emphasized that, in order to avoid potential future metabolic complications, the offspring of mothers with excessive gestational weight gain during pregnancy should be carefully monitored by their pediatricians.

## Figures and Tables

**Table 1 ijms-20-00595-t001:** Comparison of characteristics of the study subjects.

Variables	Control Group (*n* = 28)	EGWG Group (*n* = 38)	*p*
***Maternal characteristics***			
age, years	29 (24–38)	29 (28–32)	NS
pre-pregnancy BMI, kg/m^2^	20.3 (19.5–24.4)	23.2 (21.6–24.09)	NS
gestational weight gain, kg	15 (11.5–15.6)	23.9 (21–26)	**<0.001**
gestational BMI gain, kg/m^2^	5.4 (3.0–5.6)	8.4 (7.07–9.4)	**<0.001**
BMI at delivery, kg/m^2^	26.3 (24.2–29.1)	31.3 (29.7–32.05)	**<0.001**
cesarean delivery, %	14	26	NS
BMI after delivery, kg/m^2^	22 (21–23.9)	28.6 (26.2–29.7)	**<0.001**
FTI after delivery, kg/m^2^	10.1 (9.1–13.8)	14.7 (13.2–17.2)	**<0.001**
LTI after delivery, kg/m^2^	10.1. (9.4–13.1)	12.9 (11.2–13.9)	**<0.01**
***Maternal Serum***			
albumin, g/dL	3.68 (3.43–3.73)	3.55 (3.41–3.81)	NS
total cholesterol, mg/dL	249 (188–287)	225 (197–249)	NS
HDL, mg/dL	78 (75–82)	71 (59–79)	**<0.05**
LDL, mg/dL	129 (93–152)	106 (87–128)	NS
triglycerides, mg/dL	177 (150–254)	204 (178–258)	**<0.05**
HgbA1c, %	5.3 (4.6–5.4)	5.5 (5.0–5.5)	**<0.05**
SFRP5, ng/mL	3.1 (2.62–8.0)	2.47 (1.2–5.0)	**<0.0** **5**
ghrelin, ng/mL	0.933 (0.646–1.115)	1.187 (0.343–2.433)	NS
leptin, ng/mL	10.43 (6.04–14.9)	14.87 (12.6–47.6)	NS
***Umbilical Cord Blood***			
SFRP5, ng/mL	5.08 (3.74–5.69)	3.33 (2.3–4.25)	**<0.01**
ghrelin, ng/mL	0.0195 (0.187−0.282)	0.525 (0.265–1.826)	**<0.001**
leptin, ng/mL	7.53 (4.9–14.01)	10.99 (8.5–13.4)	**<0.001**
***Neonatal Anthropometric Measurements***			
birth weight, g	3630 (3200–3920)	3520 (3400–3650)	**NS**
birth body length, cm	56 (55–57)	55 (54–56)	**NS**
head circumference, cm	34 (33–35)	34 (33–35)	**NS**
chest circumference, cm	34 (34–35)	34 (33–35)	**NS**

The results are shown as the median (interquartile range: 25–75%). Statistically significant values are given in bold. BMI—body mass index; EGWG—excessive gestational weight gain; FTI—fat tissue index; HDL—high-density lipoprotein cholesterol; HgbA1c—hemoglobin A1c; LDL—low-density lipoprotein cholesterol; LTI—lean tissue index; SFRP5—secreted frizzled-related protein 5.

**Table 2 ijms-20-00595-t002:** Correlation coefficient between the umbilical cord SFRP5 levels and clinical parameters in the control and EGWG groups.

Variables	Umbilical Cord SFRP5	
	Control Group	EGWG Group
***Maternal Characteristics***		
pre-pregnancy BMI	**−0.829** ***	0.152
gestational weight gain	0.371	**−0.435** *
gestational BMI gain	0.143	**−0.442** *
BMI at delivery	**−0.6** **	−0.105
BMI after delivery	**−0.486** *	0.074
FTI after delivery	0.086	−0.342
LTI after delivery	−0.086	**0.527** **
***Maternal Serum***		
albumin	−0.143	**−0.603** **
total cholesterol	**−0.406** *	**−0.436** *
HDL	0.058	**−0.567** **
LDL	−0.371	−0.087
triglycerides	−0.058	−0.081
HgbA1c	**0.667** ***	**0.636** ***
SFRP5	**0.429** *	**0.452** *
ghrelin	**0.771** ***	−0.394
leptin	**0.6** **	−0.171
***Umbilical Cord Blood***		
ghrelin	**−0.657** ***	**−0.817** ***
leptin	−0.086	**0.495** *
***Neonatal Anthropometric Measurements***		
birth weight	−0.2	**0.781** ***
birth body length	−0.309	**0.739** ***
head circumference	−0.206	**0.532** **
chest circumference	**0.494** *	**0.516** **

Statistically significant values are given in the bold type. * *p* < 0.05; ** *p* < 0.01; *** *p* < 0.001. BMI—body mass index; EGWG—excessive gestational weight gain; FTI—fat tissue index; HDL—high-density lipoprotein cholesterol; HgbA1c—hemoglobin A1c; LDL—low-density lipoprotein cholesterol; LTI—lean tissue index; SFRP5—secreted frizzled-related protein 5.

**Table 3 ijms-20-00595-t003:** Multiple linear regression analyses for the umbilical cord SFRP5 levels.

Independent Variable	B	*β*	95% CI	*p*
maternal serum SFRP5	0.33	0.50	0.32–0.69	**<0.001**
maternal serum ghrelin	0.12	0.39	0.19−0.59	**<0.001**
umbilical cord ghrelin	−0.26	−0.79	−0.99–(−0.59)	**<0.001**
maternal serum leptin	0.06	0.72	0.52–0.92	**<0.001**
maternal LDL	−0.01	−0.23	−0.38–(−0.07)	**<0.01**
pre-pregnancy BMI	−0.12	−0.27	−0.47–(−0.06)	**<0.05**
gestational weight gain	−0.08	−0.29	−0.48–(−0.11)	**<0.01**

Adjusted for the serum SFRP5 levels, the serum and umbilical cord ghrelin and leptin levels, maternal LDL, triglycerides, HgbA1c, gestational weight gain, pre-pregnancy BMI, BMI at delivery and gestational BMI gain. Unstandardized *β* coefficients with 95% confidence interval and B linear regression coefficients are shown. Statistically significant values are given in the bold type. BMI—body mass index; LDL—low-density lipoprotein cholesterol; SFRP5—secreted frizzled-related protein 5.

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
