# Peer review of "Umbilical Cord SFRP5 Levels of Term Newborns in Relation to Normal and Excessive Gestational Weight Gain"

_ijms, 2019, doi:10.3390/ijms20030595_

Round 1

Reviewer 1 Report

Kimber-Trojnar et al. found SFRP5 concentration in cord blood to be disregulated in women with excessive weight gain in pregnancy.

In my opinion the paper is well written, the statistical manipulation of data is well performed, and conclusions are supported by the obtained results. If these data would be confirmed in a greater population and in a prospective cohort study, in my opinion the maternal-blood and umbilical cord SFRP5 concentrations could be proposed as a predictive marker for worse metabolic outcomes of the newborns.

Some minor concerns exist:

- Page 2, lines 48-62: all this section should be moved into the discussion

- Results, lines 64-65: this is a selection criterion, not a result. The sentence should be erased

- Table 1: if the asterisks are used to indicate a statistically significant difference, ar reported into the table's description, it is unuseful to report the exact p into the table

- Discussion, page 5 line 143: the weight gain average value "13.1 kg" differs from the one reported into the table 1; this latter appears to be the wrong one to me

Author Response

Dear Sir,

We would like to express our gratitude for your meaningful and helpful comments which have made a substantial contribution to the quality of our paper.

Having studied the Reviewers’ comments and following your advice, we have decided to introduce some changes into our paper, which, we do hope, will bring some improvement to our manuscript.

Our responses to the Reviewer's comments:

1.      “Page 2, lines 48-62: all this section should be moved into the discussion”

We have transferred the section (lines 48-62) from the “Introduction” in the “Discussion”.

2.      “Results, lines 64-65: this is a selection criterion, not a result. The sentence should be erased”

The sentence regarding a selection criterion has been removed: "The largest gestational weight and BMI gains were observed in the EGWG group, which was a consequence of the choice of inclusion criteria for this study group.”.

3.      “Table 1: if the asterisks are used to indicate a statistically significant difference, ar reported into the table's description, it is unuseful to report the exact p into the table”

The asterisks have been removed from the Table 1 as well as from the Table 3.

4.      “Discussion, page 5 line 143: the weight gain average value "13.1 kg" differs from the one reported into the table 1; this latter appears to be the wrong one to me”

We have modified the sentence: “Our study comprised both participants with normal pre-pregnancy BMI and different gestational weight gain; i.e. an excessive increase in body weight during pregnancy in the EGWG group (with an average value of 23.9 kilograms per pregnancy) and an appropriate gestational weight gain (with an average value of 13.1 kilograms per pregnancy) in the control group and their offspring.”

This sentence presented the average values, whereas the results shown in the Tables 1 are the median. We fully agree with the Reviewer that these additional results are not useful.

We appreciate your time and look forward to your response.

Yours faithfully,

Zaneta Kimber-Trojnar and co-authors

Reviewer 2 Report

The data presented in this paper is of interest and could potentially contribute to the understanding of the actions and significance of SFRP5 in gestational obesity. The paper aims to investigate the correlation of umbilical cord SFRP5 and metabolic risk factors in pregnancy.

Major questions

1.       Since SFRP5 is an adipokine, anthropometric indicators of fat mass (e.g. body fat percentage) should be provided if this information is available. A discussion of why SFRP5 is elevated in obesity should also be included. (i.e. Is SFRP5 elevated with increased fat mass? Association with visceral or subcutaneous fat deposition?)

2.       Line 60: “EGWG may expose the developing foetus to persistently raised concentrations of glucose, insulin, amino acids, and lipids as well as imbalance between pro- and anti-inflammatory adipokines derived from maternal adipose tissue.”

This needs references.

3.       Figure 1 is unnecessary since the data is already presented in Table 1.

4.       Line 226: “Secondly, it appears of great clinical significance to monitor the circulating SFRP5 levels in all trimesters of pregnancies as well as in future life of children.”

How is this a limitation?

Minor points:

1.       Please check the whole manuscript for consistent use of decimals (use only the decimal points and refrain from using commas).

2.       Standard significance levels are * p<0.05; ** p<0.01; ***p<0.001. Where P-values are less than 0.001, values should be expressed as P<0.001 or in expressed in exponential notation.

3.       Italicise gene names e.g. Sfrp5

4.       ‘Periparturient’ is misspelled as ‘peri-parturient’.

5.       Line 163: Use ‘increased’ in place of ‘strengthened’.

6.       Lines 216 to 224 should be incorporated in the discussion of study limitations.

Author Response

Responses to Reviewer's Comments (Reviewer 2)

Dear Sir,

We would like to express our gratitude for your meaningful and helpful comments which have made a substantial contribution to the quality of our paper.

Having studied the Reviewers’ comments and following your advice, we have decided to introduce some changes into our paper, which, we do hope, will bring some improvement to our manuscript.

Our responses to the Reviewer's comments:

Major questions

1.        “Since SFRP5 is an adipokine, anthropometric indicators of fat mass (e.g. body fat percentage) should be provided if this information is available. A discussion of why SFRP5 is elevated in obesity should also be included. (i.e. Is SFRP5 elevated with increased fat mass? Association with visceral or subcutaneous fat deposition?)”

SFRP5 is an anti-inflammatory adipokine that regulates metabolic homeostasis. SFRP5 as an inhibitor of Wnt signaling has been reported to be implicated in obesity, insulin resistance, dyslipidemia and metabolic syndromes. As suggested we have added additional results regarding maternal body composition, i.e. lean and fat tissue indexes.

We have modified one sentence in the “Abstract”:

“Umbilical cord SFRP5 concentrations were directly associated with the maternal serum SFRP5, and haemoglobin A1c and lean tissue index, umbilical cord leptin levels as well as with newborns’ anthropometric measurements in the EGWG subjects.”

We have modified two sentences in the “Results”:

“The EGWG women were also characterized by increased levels of haemoglobin A1c (HgbA1c), and triglycerides and indexes of fat (FTI) and lean (LTI) tissues and as well as lower concentrations of high-density lipoprotein cholesterol (HDL).”

“In the EGWG group we observed a direct correlation between the umbilical cord SFRP5 and the maternal serum HgbA1c, and SFRP5 and LTI after delivery, the umbilical cord leptin levels as well as the with all the four newborns’ anthropometric measurements (i.e. with neonatal birth weight, birth body length, head and chest circumference).”

We have added the part concerning associations between SFRP5 and obesity to the “Discussion”:

“The expression of Sfrp5 is slowly induced in the process of differentiation of white and brown adipocytes and increased in mature adipocytes [47]. Lower SFRP5 levels have been detected in obese subjects in contrast with lean subjects in studies analysing correlations between SFRP5 and adiposity indicators such as BMI, waist-hip ratio, body fat percentage and lipid profile [9]. In our study a positive correlation was found between the umbilical cord serum SFRP5 concentrations and maternal LTI, but only in the EGWG group. LTI, which is defined as the lean tissue mass divided by the square of the body height, expresses the muscle mass [48]. Mori et al. [49] revealed that SFRP5 promotes adipocyte growth by repressing Wnt signaling and decreasing oxidative metabolism as an endogenous suppressor of adipogenesis. Rulifson et al. [50] and Van Camp et al. [51] found that a significant increase in Sfrp5 expression was observed amongst adipose tissues in obese mice and genetic variation in Sfrp5 could determine the distribution and volume of both subcutaneous and abdominal fat in obese males, respectively.”

We have added one sentence to the “Materials and Methods”:

“Maternal body composition (LTI and FTI) were evaluated by the bioelectrical impedance analysis (BIA) method with the use of a body composition monitor (BCM) (Fresenius Medical Care) in the early post-partum period (i.e. 48 hours after delivery).”

We have also added five references and changed the order of references.

47.       Wang, R.; Hong, J.; Liu, R.; Chen, M.; Xu, M.; Gu, W.; Zhang, Y.; Ma, Q.; Wang, F.; Shi, J.; Wang, J.; Wang, W.; Ning, G. SFRP5 acts as a mature adipocyte marker but not as a regulator in adipogenesis. J Mol Endocrinol. 2014, 53, 405-415. doi: 10.1530/JME-14-0037.

48.       Wang, Y.W.; Lin, T.Y.; Peng, C.H.; Huang, J.L.; Hung, S.C. Factors Associated with Decreased Lean Tissue Index in Patients with Chronic Kidney Disease. Nutrients 2017, 9, pii: E434. doi: 10.3390/nu9050434.

49.       Mori, H.; Prestwich, T.C.; Reid, M.A.; Longo, K.A.; Gerin, I.; Cawthorn, W.P.; Susulic, V.S.; Krishnan, V.; Greenfield, A.; Macdougald, O.A. Secreted frizzled-related protein 5 suppresses adipocyte mitochondrial metabolism through WNT inhibition. J Clin Invest. 2012, 122, 2405-2416. doi: 10.1172/JCI63604.

50.       Rulifson, I.C.; Majeti, J.Z.; Xiong, Y.; Hamburger, A.; Lee, K.J.; Miao, L.; Lu, M.; Gardner, J.; Gong, Y.; Wu, H.; Case, R.; Yeh, W.C.; Richards, W.G.; Baribault, H.; Li, Y. Inhibition of secreted frizzled-related protein 5 improves glucose metabolism. Am J Physiol Endocrinol Metab. 2014, 307, E1144-E1152. doi: 10.1152/ajpendo.00283.2014. 

51.       Van Camp, J.K.; Beckers, S.; Zegers, D.; Verrijken, A.; Van Gaal, L.F.; Van Hul, W. Common genetic variation in sFRP5 is associated with fat distribution in men. Endocrine 2014, 46, 477-484. doi: 10.1007/s12020-013-0088-7.

2.       Line 60: “EGWG may expose the developing foetus to persistently raised concentrations of glucose, insulin, amino acids, and lipids as well as imbalance between pro- and anti-inflammatory adipokines derived from maternal adipose tissue.” This needs references.

We have added two references regarding the above listed sentence:

Estampador, A.C.; Pomeroy, J.; Renström, F.; Nelson, S.M.; Mogren, I.; Persson, M.; Sattar, N.; Domellöf, M.; Franks, P.W. Infant body composition and adipokine concentrations in relation to maternal gestational weight gain. Diabetes Care 2014, 37, 1432-1438. doi: 10.2337/dc13-2265.

Walsh, J.M.; McGowan, C.A.; Mahony, R.M., Foley, M.E., McAuliffe, F.M. Obstetric and metabolic implications of excessive gestational weight gain in pregnancy. Obesity (Silver Spring) 2014, 22, 1594-1600 doi:10.1002/oby.20753.

3.       Figure 1 is unnecessary since the data is already presented in Table 1.

We have removed Figure 1.

4.       Line 226: “Secondly, it appears of great clinical significance to monitor the circulating SFRP5 levels in all trimesters of pregnancies as well as in future life of children.” How is this a limitation?

We have modified the article part regarding the study limitations:

“On the other hand, Tthis study has two an important limitations. limitation as Firstly, this is a small sample study. Our results require further verification. Secondly, it It also appears of great clinical significance to monitor the circulating SFRP5 levels in all trimesters of pregnancies as well as in future life of children.”

Minor points:

1.       Please check the whole manuscript for consistent use of decimals (use only the decimal points and refrain from using commas).

We have corrected decimals with using commas in the whole article.

2.       Standard significance levels are * p<0.05; ** p<0.01; ***p<0.001. Where P-values are less than 0.001, values should be expressed as P<0.001 or in expressed in exponential notation.

We have corrected statistically significant differences in the table's description as:

* p<0.05; ** p<0.001; ***p<0.0001. P-values less than 0.001 have been expressed as P<0.001.

As suggested by Reviewer 1 the asterisks have been removed from the Table 1 and Table 3.

3.       Italicise gene names e.g. Sfrp5

We have applied the italic font version for gene names.

4.       ‘Periparturient’ is misspelled as ‘peri-parturient’.

We have corrected the indicated word: “Our results revealed differences at the periparturient period not only in the gestational weight and BMI gains between these two groups of mothers but in the laboratory results as well.”

5.       Line 163: Use ‘increased’ in place of ‘strengthened’.

We have modified the sentence: “Christodoulides et al. [39] reported that Sfrp5 mediated epigenetic silencing of the Wnt signaling pathway in the white adipose tissue could lead to a strengthened an increased adipogenesis with a significant likelihood of increasing susceptibility to diet-induced obesity in the mice models.”

6.     Lines 216 to 224 should be incorporated in the discussion of study limitations.

This section was combined with study limitations.

We appreciate your time and look forward to your response.

Yours faithfully,

Zaneta Kimber-Trojnar and co-authors
